# Measuring the experience of social connection within specific social interactions: The Connection During Conversations Scale (CDCS)

**Karynna Okabe-Miyamoto**[1]*, **Lisa C. Walsh**[2], **Daniel J. Ozer**[1], **Sonja Lyubomirsky**[1]

**1** Department of Psychology, University of California, Riverside, Riverside, California, United States of America, **2** Department of Psychology, University of California, Los Angeles, Los Angeles, California, United States of America

* Karynnaom@gmail.com

## Abstract

Decades of research have demonstrated that social connection is fundamental to health and well-being. The benefits of connection are observed with both close and distant others, within both new and established relationships, and even with exchanges that unfold over a relatively short timeframe. Because social connection is fundamental to well-being, many existing measures in the literature aim to assess either a global sense of connection or partner-specific (relationship-specific) connection. What is missing are measures of connection felt in specific social interactions or conversations. In three studies (Study 1: $N = 351$; Study 2: Time 1 $N = 397$, Time 2 $N = 336$, Time 3 $N = 299$; Study 3: $N = 235$), we developed the Connection During Conversations Scale (CDCS), a 14-item measure of conversation-specific social connection that assesses connection experienced during a social interaction (or conversation). Confirmatory factor analyses demonstrated that a four-factor model fit our samples well, which resulted in four subscales: Shared Reality, Partner Responsiveness, Participant Interest, and Affective Experience. The overall CDCS measure, along with its four subscales, was significantly correlated with established measures of loneliness, partner responsiveness, relatedness, positivity resonance, and shared reality. Because of the importance of frequent interactions—whether with family, friends, coworkers, or strangers—our new scale will allow researchers to better understand how, when, and where such conversations may contribute to social connection and well-being. (225 words).

## Introduction

Social connection (or belonging) is essential for optimal human functioning [1, 2]. A great deal of evidence has demonstrated that social connection is associated with well-being [3–5], and that lack of social connection is a major health risk factor [6–9]. According to self-determination theory [10], relatedness (i.e., connectedness)—along with competence and autonomy—is one of the three basic psychological needs that, when fulfilled, promotes well-being.

**Data Availability Statement:** Data and R code are available at: https://osf.io/ns5mv/.

**Funding:** The authors received no specific funding for this work.

**Competing interests:** The authors have declared that no competing interests exist.

Additional research indicates that people with extraverted personalities experience greater happiness than introverts, and that engaging in extraverted behaviors (such as socially interacting with others) can enhance well-being [11]. Taken together, research over the past several decades has revealed the fundamental nature of relationships for human health and well-being.

Social connection can be defined as the experience of feeling close and connected to others, encompassing a sense of belonging, attachment, and interpersonal relationships [1, 8, 12, 13]. It includes the quality and quantity of social interactions, as well as the subjective experience of being connected to others. Using a variety of methodologies, a large literature has explored the well-being outcomes associated with feelings of social connection, as well as the specific constructs and facets (e.g., partner responsiveness, shared positive affect) that may compose connecting experiences. For example, in a longitudinal study that followed married or cohabiting couples over the course of 10 years, partner responsiveness (that is, feeling understood, valued, and cared for) predicted greater eudaimonic well-being [14]. During the stressful transition into parenthood, parents who reported stronger social support were less depressed during the transition period [15]. Moreover, using the Day Reconstruction Method, participants who reported greater perceived positivity resonance (i.e., shared positive affect and mutual concern) with their interaction partner, also reported greater flourishing mental health [16]. As such, correlational evidence suggests that social connection is related to beneficial well-being outcomes.

In addition to correlational studies, experimental work has also explored the link between social connection and well-being. In a study of prosocial spending, those who gave away a gift card were happier than those who kept the gift card for themselves, with the greatest well-being benefits for individuals who reported feeling connected with their gift card recipient [17]. These results demonstrate that social connection can be leveraged to develop or strengthen happiness-boosting interventions. Importantly, individuals instructed to engage socially report relatively more connectedness and positive emotion [11, 18, 19]. Overall, these studies support the notion that stronger self-reported feelings of social connection—assessed and induced in a variety of ways—are related to myriad well-being outcomes throughout the lifespan and during major life transitions.

In addition to well-being outcomes, social connection has also been associated with positive physical health and improved cognitive outcomes. In a meta-analysis of 148 studies, researchers found that individuals who reported having relatively stronger social relationships, regardless of age or gender, had a 50% greater likelihood of surviving than those without strong relationships [8]. In the longitudinal study of married or cohabiting couples, partner responsiveness also predicted healthier cortisol levels at a 10-year follow-up [20]. In one experiment that administered mock personality tests then provided false personality feedback, participants who were told that they would have meaningful relationships in the future performed better on verbal, math, and spatial assessments compared to those told that they would end up alone later in life [21]. As such, induced feelings of social connection are not only linked to improved well-being outcomes but improved cognitive functioning as well.

What is it about felt social connection that facilitates well-being? To investigate this question, researchers have begun to use varied methodologies to probe people's social interactions. For example, both self-report and audio recording data using the Electronically Activated Recorder (EAR) have shown that engaging in more conversations is related to greater well-being [22–24]. Furthermore, people who connect through conversations report to be happier than those who do not, whether those conversations are with close others [25] or strangers (e.g., baristas, bus strangers) [26, 27]. Moreover, both engaging in a relatively larger number of conversations and having deeper (versus small talk) social interactions have been found to be

related to greater well-being [28]. Interestingly, the quantity and quality of social interactions may be valued differently depending on one's age, such that individuals in their 20s may prefer quantity while those in their 30s may prefer quality [29, 30]. Therefore, not only are the number of conversations important for well-being, but so is their quality—or sense of connection or understanding they provide.

In sum, a number of correlational and experimental studies have provided evidence that social interactions and conversations, with both close others and strangers, are associated with greater happiness [11, 18, 19, 25–27, 31]. However, little is known about how connected people feel during these conversations or interactions. Most research on social connection relies on either of two approaches to assess felt social connection: global relationship measures (e.g., how satisfied someone feels with the amount and quality of their social connection across all connections) and specific partner measures (e.g., how satisfied someone feels with their connection with a spouse, friend, sister, etc.). An alternative approach might examine the degree of social connection experienced in a specific social interaction (e.g., how satisfied someone feels with their connection during or after a particular conversation). In other words, research is needed to assess the quality of social connection moments, such as a phone call with a parent or a chat with a co-worker. Whether a particular conversation is lengthy or hasty, it has the capacity to influence how connected people feel. Indeed, brief interactions with weak ties, such as chats with baristas or Lyft drivers, have been shown to lead to feelings of social connection and well-being [32], and almost all interpersonal relationships essentially comprise a series of multiple social interactions. Accordingly, it is imperative to possess tools to advance understanding of how individual social interactions influence social connection and well-being. Before we introduce such a tool in this paper, we first briefly review a selection of measures of social connection previously used in the literature.

## Existing measures of social connection

Social connection can be explored at multiple levels, ranging from global (e.g., "Do you feel a sense of intimacy and closeness with others?") to partner-specific (e.g., "Do you feel close to your" husband or parent) to interaction-specific (e.g., "Did you feel a sense of connection during this conversation?"). As a result, many existing relevant measures—global and partner-specific ones, in particular—can be found in the literature. We outline several representative measures below. Additionally, we present a full list and description of all existing measures identified in Supplemental Materials (see S1 Table).

**Global relationship measures.** Global relationship measures typically ask respondents to holistically evaluate their relationships (see Global section of S1 Table). For example, the Social Provisions Scale [33] includes items like "There are people I can depend on to help me if I really need it," and the Balanced Measure of Psychological Needs (BMPN) [34] has items like "I felt close and connected with other people who are important to me." Similar measures include the Multi-Dimensional Scale of Perceived Social Support (MSPSS) [35], with items such as "There is a special person with whom I can share my joys and sorrows"; the Social Connectedness Scale [36], with items like "I feel understood by the people I know"; and the support ("There are people who give me support and encouragement") and belonging ("I feel a sense of belonging in my community") subscales of the Comprehensive Inventory of Thriving (CITI) [37]. The UCLA Loneliness scale [38] assesses the general lack of connection, or feelings of loneliness, with items such as "No one really knows me well." These measures are critical vis-à-vis their ability to tap into how much connectedness an individual feels in general. However, they were not designed to examine the strength of a person's connection in specific relationships or during specific conversations.

**Partner-specific relationship measures.** Another category of connection measures asks individuals about the connection they feel with a specific partner (see Partner-Specific section of S1 Table). One type of partner-specific relationship measure assesses the connection people feel *from* their relationship partner. Examples include the Partner Responsiveness Scale [39, 40], with items such as "Compared to most experiences I've had meeting somebody new, I get the feeling that this person sees the 'real' me'"; and the Relationship Closeness Inventory [41], with items such as "[My partner] influences important things in my life." Another type of partner-specific relationship measure assesses the connection people feel *toward* their relationship partner. Such measures include the Relationship Assessment Scale [42] with items like, "How much do you love your partner?" Finally, some scales measure both connection people feel *from* and *toward* their relationship partner. These include the Inclusion of Other in the Self Scale [43], which displays seven options involving two circles that range from separate to increasingly close (and eventually overlapping) to tap perceived closeness between self and partner; and the Two-Way Social Support Scale [44], with items like, "I am there to listen to others' problems" and "There is someone I can talk to about the pressures in my life." In sum, these three types of partner-specific measures allow researchers to examine connection with a specific partner, but they do not capture people's sense of connection during a specific social interaction.

**Interaction-specific connection.** Despite research evidence demonstrating links between well-being and the frequency of social interactions, to our knowledge, only two measures—both recently developed—gauge the amount or quality of connection felt during a particular social interaction. The Positivity Resonance Scale [16] asks respondents questions such as, what percentage of time (from 0 to 100%) "Did you feel 'in sync' with the other(s)?" among other questions about several features of an interaction. Motivated by the theory of positivity resonance, this 7-item scale aims to measure its three hypothesized facets—namely, shared positive affect, mutual care and concern, and behavioral and biological synchrony [45], with some items tapping more than one facet. However, when assessing social connection felt during an interaction, the Positivity Resonance Scale might miss important features of the interaction, such as general affective experience. Furthermore, because the scale follows the positivity resonance theory, this might be limiting, such that connection might be found not only during shared positive affect but also shared negative affect (e.g., shared misery). Additionally, respondents have reported that percent conversation time from 0 to 100 is complicated to estimate accurately, potentially making the scale relatively time consuming and cognitively taxing [46]. As such, a different measure may be needed to assess types of connecting experiences that may not cover all three of these elements or feature additional elements.

The Generalized Shared Reality Measure [47], published after our data collection had completed, is another interaction-specific measure of connection that can be used for both close others and strangers. It includes items such as "during our interaction we thought of things at the same time." However, this measure, which is also theoretically motivated, is designed to focus on only one facet of social connection—namely, shared reality. Overall, the literature is still missing a scale that more broadly assesses felt social connection during specific social interactions which greatly limits the study of social connection. For example, in order to create interventions to help people connect, researchers must understand how people connect in daily conversations. A measure of connection felt during an interaction can help researchers understand what aspects of conversations make for the most connecting experiences (e.g., commonalities). Thus, researchers can identify strategies to target these key aspects of conversations in order to boost overall connection (e.g., arming people with questions to ask others that might reveal commonalities). Additionally, a measure of connection felt during an interaction may allow researchers to identify profiles of those struggling to connect with others,

such as those who have trouble finding commonalities with others or those who view all interactions in a negative light. By identifying these profiles, researchers may more easily create overarching strategies to help people who fall under different social connection profiles.

## The present studies

Our aim was to create a measure of social connection to assess connection felt during interactions or conversations with both close others and strangers in daily life. To this end, we conducted a set of programmatic studies to develop and validate the Connection During Conversations Scale. For Study 1, we collected a broad pool of items from existing measures of social connection, including the Positivity Resonance Scale [16], the Partner Responsiveness Scale [39, 40] and the Social Provisions Scale [33], to create our new scale (see S1 Table in Supplemental Materials for a full list and description of these existing measures). Next, we evaluated our new measure—the 16-item version in Studies 2a, 2b, and 2c and the 14-item version in Study 3—by correlating it with the most commonly used and most relevant social connection measures in the literature and provided construct validity evidence by examining correlations with personality, well-being, and demographic variables.

## Study 1

Our first study focused on creating the Connection During Conversations Scale (CDCS), designed to be a measure of social connection felt during a specific interpersonal interaction. Based on a comprehensive search of social connection scales (again, see S1 Table for a full list of scales used to develop the CDCS), we selected 53 items (i.e., items that were the most relevant to social connection, adapted, and edited for clarity) to construct our new measure. Additionally, to ensure strong recall and deep reflection of a recent social interaction, we also created an open-ended prompt that asked participants to write about this interaction. Following the prompt, participants completed the 53-item measure, then provided details about where, when, and with whom the interaction occurred.

### Method

**Participants.**   Participants ($N$ = 351) were recruited from Prolific Academic based on available department funding, an online platform used to recruit subjects that has been shown to provide good quality online data [48]. To join the study, they had to be fluent in English and have an "approval rating" of over 90% on Prolific. Participants ranged in age from 18 to 66 ($M$ = 30.92, $SD$ = 10.12). They were mostly male (56%) and Caucasian (63%), and nearly half were single and never married (42%). They also resided around the world, with 34% from the U.S., 17% from the U.K., 11% from Canada, and the remaining 38% from 26 other international countries (e.g., Australia, South Korea).

**Procedure.**   Participants joined a 15-min study entitled "Social Interaction Psychological Research Study." Following written consent online, they completed our writing prompt, which asked them to take a few moments to describe a social interaction that had taken place within the last 2 days:

> For the next few minutes, think about *a recent interaction or conversation* you had with another (one) person that lasted for at least a few moments. . .Now, we would like you to briefly describe this interaction. . .What happened during the interaction or conversation? What were you thinking and/or feeling during the interaction? Where were you?

The resulting qualitative data from this prompt are beyond the scope of the present study and are not presented here. Next, participants completed our 53-item connection measure, followed by questions about their target social interaction (e.g., whether the interaction was positive, negative, or neutral), their interaction partner (e.g., how long they had known them), and demographic items (e.g., their own age, gender). Participants who completed the study were compensated $2.00 for their time.

## Materials

**Connection scale item pool.** To compile a pool of items, we turned to reliable and valid scales already published in the empirical literature that aim to assess aspects of social connection and interpersonal relationships (again, see S1 Table for existing scales used in scale creation). While examining each measure, we identified items that were most closely aligned with social connection during conversations, resulting in a pool of 53 items. Furthermore, we modified and updated some items for clarity (e.g., removed or separated double-barreled questions). Of the 53 items, 33 were categorized as being toward one's partner (e.g., "I felt 'in sync' with them"), 15 were categorized as being from one's partner (e.g., "They were responsive to me"), and 5 were categorized as being general items (e.g., "The interaction brightened my day"). Each of the items within each category was presented together with blocks counterbalanced and items within the blocks randomized. Each category was presented in separate blocks to reduce participant burden, as switching between these types of questions could increase cognitive load. Participants rated their level of agreement with each item on a 1 (*strongly disagree*) to 7 (*strongly agree*) Likert scale.

**Interaction partner demographics and interaction details.** We asked participants to respond to several questions about their interaction partners, including their partner's gender, age, ethnicity, how long the participant has known their partner (ranging from just met to many years), and who their partner was (e.g., close friend, brother/sister, stranger). We also asked participants to indicate when the interaction occurred, its mode of communication, and the interaction's duration and valence. See Table 1 for a breakdown of demographics and details for this study (as well as Study 2 and 3).

## Results

**Exploratory factor analysis.** To determine the number of factors that emerged from our 53 items, we calculated eigenvalues for each of our dimensions and then graphed the eigenvalues (Eigenvalues: 25.88, 2.16, 2.07, 1.89, 1.39, 1.31, 1.15, 0.96) using a scree plot. It appeared that 2 or 4 factors may be present in our data. Based on Horn's Parallel Analysis for component retention using 5000 iterations, 4 components were retained. Each of the 4 components contained 4 items, for a total of 16 retained items. Therefore, we decided to extract four factors with our data. We fit the four-factor model to our data using the *fa* function in the psych package in R. We used the maximum likelihood method with oblimin rotation (because we expected our factors to be correlated), which resulted in a solution that accounted for 57% of the cumulative variance.

The 16 retained items were correlated (average inter-item $r = .54$). The four factors were also correlated (average $r = .63$). The correlation between the Shared Reality latent variable was stronger with the Partner Responsiveness latent variable ($r = .70$) and the Participant Interest latent variable ($r = .51$) than with the Affective Experience latent variance ($r = -.49$). The Partner Responsiveness latent variable were oppositely correlated with the Participant Interest latent variable ($r = .52$) and the Affective Experience latent variable ($r = -.50$). Finally, the

**Table 1. Partner demographics and interaction details.**

| | Study 1 | | Study 2 | |
| --- | --- | --- | --- | --- |
| | | Time 1 | Time 2 | Time 3 |
| **Interaction Partner Demographics** | | | | |
| Gender | 47% Male | 49% Male | 46% Male | 48% Male |
| | 52% Female | 51% Female | 54% Female | 52% Female |
| | 1% Nonbinary | 0% Nonbinary | 0% Nonbinary | 0% Nonbinary |
| | < 1% Unknown | 0% Unknown | 0% Unknown | 0% Unknown |
| Age | $M = 35.74, SD = 15.88$ | $M = 35.28, SD = 15.63$ | $M = 38.04, SD = 6.71$ | $M = 38.70, SD = 15.73$ |
| | Range: 14–87 | Range: 8–92 | Range: 6–86 | Range: 8–82 |
| Ethnicity | 0% Native American/Alaskan | < 1% Native American/Alaskan | < 1% Native American/Alaskan | < 1% Native American/Alaskan |
| | 14% Asian | 8% Asian | 7% Asian | 8% Asian |
| | 3% Black/African American | 5% Black/African American | 3% Black/African American | 4% Black/African American |
| | 0% Hawaiian/Pacific Islander | 0% Hawaiian/Pacific Islander | 1% Hawaiian/Pacific Islander | < 1% Hawaiian/Pacific Islander |
| | 66% White/Caucasian | 77% White/Caucasian | 79% White/Caucasian | 78% White/Caucasian |
| | 11% Hispanic/Latino | 5% Hispanic/Latino | 5% Hispanic/Latino | 5% Hispanic/Latino |
| | 2% Middle Eastern | 2% Middle Eastern | 1% Middle Eastern | 1% Middle Eastern |
| | 1% More Than One | 1% More Than One | 1% More Than One | 1% More Than One |
| | 1% Other | 1% Other | 1% Other | 1% Other |
| | 3% Unknown | < 1% Unknown | 2% Unknown | 1% Unknown |
| How Long Have You Known Your Interaction Partner | 11% We Just Met | 9% We Just Met | 9% We Just Met | 8% We Just Met |
| | 2% A Few Hours | 1% A Few Hours | < 1% A Few Hours | 1% A Few Hours |
| | 1% A Few Days | 2% A Few Days | 1% A Few Days | 1% A Few Days |
| | 4% A Few Weeks | 3% A Few Weeks | 2% A Few Weeks | 2% A Few Weeks |
| | 9% A Few Months | 10% A Few Months | 4% A Few Months | 4% A Few Months |
| | 10% About A Year | 9% About A Year | 7% About A Year | 6% About A Year |
| | 26% A Few Years | 25% A Few Years | 18% A Few Years | 19% A Few Years |
| | 36% Many Years | 42% Many Years | 57% Many Years | 59% Many Years |
| Who Is Your Interaction Partner? | 11% Stranger | 8% Stranger | 9% Stranger | 7% Stranger |
| | 7% Acquaintance | 7% Acquaintance | 4% Acquaintance | 5% Acquaintance |
| | 12% Casual (Non-Romantic) Friend | 12% Casual (Non-Romantic) Friend | 7% Casual (Non- Romantic) Friend | 11% Casual (Non- Romantic) Friend |
| | 18% Close (Non-Romantic) Friend | 25% Close (Non-Romantic) Friend | 24% Close (Non-Romantic) Friend | 23% Close (Non-Romantic) Friend |
| | 11% Parent | 11% Parent | 18% Parent | 19% Parent |
| | 3% Child | 2% Child | 1% Child | 1% Child |
| | 4% Brother/Sister | 4% Brother/Sister | 7% Brother/Sister | 7% Brother/Sister |
| | 1% Grandparent | 1% Grandparent | 1% Grandparent | 0% Grandparent |
| | < 1% Aunt/Uncle | 1% Aunt/Uncle | 1% Aunt/Uncle | 1% Aunt/Uncle |
| | 7% Coworker | 8% Coworker | 5% Coworker | 5% Coworker |
| | 4% Boss/Supervisor | 1% Boss/Supervisor | 2% Boss/Supervisor | 2% Boss/Supervisor |
| | < 1% Someone You Supervise | 1% Someone You Supervise | < 1% Someone You Supervise | 0% Someone You Supervise |
| | 1% Professor/TA | 1% Professor/TA | < 1% Professor/TA | 0% Professor/TA |
| | 6% Husband/Wife | 5% Husband/Wife | 6% Husband/Wife | 6% Husband/Wife |
| | 8% Serious Relationship partner | 7% Serious Relationship Partner | 8% Serious Relationship Partner | 8% Serious Relationship Partner |
| | 1% Casual Relationship Partner | 1% Casual Relationship Partner | 1% Casual Relationship Partner | 1% Casual Relationship Partner |
| | 1% New Romantic Partner | 1% New Romantic Partner | 2% New Romantic Partner | 1% New Romantic Partner |
| | 5% Other | 6% Other | 4% Other | 4% Other |
| **Interaction Details** | | | | |
| When Did the Interaction Occur? | 46% Today | 48% Today | 57% Today | 31% Today |
| | 50% Yesterday | 47% Yesterday | 71% Yesterday | 63% Yesterday |
| | 4% Other | 5% Other | 6% Other | 6% Other |

(*Continued*)

**Table 1.** (Continued)

| | Study 1 | | Study 2 | |
|---|---|---|---|---|
| | | Time 1 | Time 2 | Time 3 |
| *Where Did the Interaction Occur?* | 65% Face-to-Face | 63% Face-to-Face | 41% Face-to-Face | 46% Face-to-Face |
| | 10% Phone (Audio) | 14% Phone (Audio) | 20% Phone (Audio) | 21% Phone (Audio) |
| | 2% Video Chat | 4% Video Chat | 15% Video Chat | 12% Video Chat |
| | 14% Text | 10% Text | 13% Text | 8% Text |
| | 4% Social Media | 5% Social Media | 6% Social Media | 8% Social Media |
| | 5% Other | 4% Other | 4% Other | 5% Other |
| *How Long Was the Interaction?* | 19% ≤ 5 mins | 19% ≤ 5 mins | 16% ≤ 5 mins | 13% ≤ 5 mins |
| | 46% 5–30 mins | 49% 5–30 mins | 52% 5–30 mins | 48% 5–30 mins |
| | 15% 30 mins– 1 hour | 14% 30 mins– 1 hour | 20% 30 mins– 1 hour | 17% 30 mins– 1 hour |
| | 11% 1–2 hours | 10% 1–2 hours | 6% 1–2 hours | 13% 1–2 hours |
| | 5% 2–3 hours | 5% 2–3 hours | 2% 2–3 hours | 6% 2–3 hours |
| | 2% 3–4 hours | 2% 3–4 hours | 1% 3–4 hours | 2% 3–4 hours |
| | 1% 4–5 hours | 1% 4–5 hours | 1% 4–5 hours | 0% 4–5 hours |
| | 1% 5+ hours | 1% 5+ hours | 1% 5+ hours | 1% 5+ hours |
| *Valence* | *M* = 5.42, *SD* = 1.51 | *M* = 5.47, *SD* = 1.59 | *M* = 5.10, *SD* = 1.66 | *M* = 5.27, *SD* = 1.65 |
| | 2% Rated as a 1 | 2% Rated as a 1 | 2% Rated as a 1 | 4% Rated as a 1 |
| | 5% Rated as a 2 | 7% Rated as a 2 | 9% Rated as a 2 | 5% Rated as a 2 |
| *1 = Negative* | 6% Rated as a 3 | 6% Rated as a 3 | 9% Rated as a 3 | 6% Rated as a 3 |
| *4 = Neutral* | 11% Rated as a 4 | 10% Rated as a 4 | 13% Rated as a 4 | 14% Rated as a 4 |
| *7 = Positive* | 15% Rated as a 5 | 13% Rated as a 5 | 14% Rated as a 5 | 9% Rated as a 5 |
| | 37% Rated as a 6 | 31% Rated as a 6 | 33% Rated as a 6 | 39% Rated as a 6 |
| | 25% Rated as a 7 | 32% Rated as 7 | 21% Rated as a 7 | 23% Rated as a 7 |

Participant Interest latent variable and the Affective Experience latent variable were also negatively correlated (*r* = -.37).

Furthermore, the items within each of the four factors appeared to cluster in ways that represented meaningful constructs in the literature (e.g., partner responsiveness). To determine the final items within each of our four factors, we first removed items that loaded below .50. If items were semantically similar, the item with the highest factor loading was chosen (e.g., "they respected my beliefs and opinions" over "they valued my beliefs and opinions"). Based on these criteria, 16 final items were chosen (4 items in each factor; see Table 2 for factor loadings). The final four-factor structure closely represents four constructs found in the literature to be theoretically related to social connection: (1) Shared Reality, (2) Partner Responsiveness, (3) Participant Interest, and (4) Affective Experience.

**Confirmatory factor analysis.** Next, we conducted a confirmatory factor analysis (CFA) using the *cfa* function in the lavaan package in R based on our 16-item measure of connection (4 items for each of our 4 subscales) to determine whether our four-factor solution was a good fit. A four-factor CFA fit our connection items well, $\chi^2(98)$ = 336.84, CFI = .933, TLI = .918, RMSEA = .083, 90% CI [.074, .093], SRMR = .054 (see Table 2 for factor loadings).

The 16-items of the CDCS were correlated (average inter-item *r* = .54). The four subscales of this scale were also correlated (average *r* = .63). Correlations among latent variables were strong. The Shared Reality latent variable was strongly correlated with the Partner Responsiveness latent variable (*r* = .86), the Participant Interest latent variable (*r* = .76), and the Affective Experience latent variance (*r* = -.78). The Partner Responsiveness was also strongly correlated with the Participant Interest latent variable (*r* = .77) and the Affective Experience latent

**Table 2. Items and factor loadings (Study 1, 2, and 3).**

| | | | Study 1 | Study 2 | | | Study 3 |
|---|---|---|---|---|---|---|---|
| | | | | Time 1 | Time 2 | Time 3 | |
| | | *N* | 351 | 397 | 336 | 299 | 235 |
| | | Mean | 5.24 | 5.40 | 5.51 | 5.48 | 5.13 |
| | Standard Deviation | | 1.08 | 1.11 | 1.08 | 1.16 | .95 |
| | | Alpha | .93 | .93 | .93 | .95 | .91 |
| | | Omega | .95 | .95 | .95 | .96 | .94 |
| | Item | Factor | | | Loadings | | |
| 1 | I felt "in sync" with them | SR | .91 | .86 | .91 | .90 | .78 |
| 2 | I felt like we shared a lot in common | SR | .85 | .84 | .87 | .89 | .80 |
| 3 | I felt that we saw the world in the same way | SR | .82 | .83 | .85 | .88 | .55 |
| 4 | They were able to relate to my experiences | SR | .77 | .80 | .84 | .84 | .75 |
| 5 | They were interested in my thoughts and feelings | PR | .86 | .87 | .88 | .85 | .78 |
| 6 | They respected my beliefs and opinions | PR | .81 | .82 | .84 | .88 | .75 |
| 7 | I felt that they cared about me | PR | .80 | .85 | .87 | .79 | .74 |
| 8 | They really understood who I am | PR | .80 | .84 | .80 | .85 | .78 |
| 9 | I was truly attentive during the interaction | PI | .62 | .54 | .57 | .64 | .64 |
| 10 | I was interested in their thoughts and feelings | PI | .79 | .70 | .80 | .81 | .68 |
| 11 | I thought that they were boring (R) | PI | -.75 | -.75 | -.69 | -.79 | -.70 |
| 12 | I was distracted during the conversation (R) | PI | -.52 | -.47 | -.42 | -.55 | - |
| 13 | I was nervous during the interaction (R) | AE | .57 | .44 | .37 | .47 | - |
| 14 | I felt that my energy was drained by the interaction (R) | AE | .71 | .74 | .74 | .80 | .75 |
| 15 | I couldn't wait for the interaction to end (R) | AE | .81 | .81 | .82 | .86 | .75 |
| 16 | I felt that it was hard to communicate with them (R) | AE | .81 | .80 | .78 | .84 | .57 |

*Note.* SR = Shared Reality factor. PR = Partner Responsiveness factor. PI = Participant Interest factor. AE = Affective Experience factor. The items used in Study 3 are the final 14-items in our measure.

variable (*r* = -.72). Finally, the Participant Interest latent variable and the Affective Experience latent variable were also negatively correlated (*r* = -.80). Notably, Affective Experience was negatively correlated with the three other latent variables (Shared Reality, Participant Interest, and Partner Responsiveness).

## Brief discussion

In Study 1, we developed a16-item, four-factor measure. In Study 2, we aimed to evaluate this 16-item interaction-specific social connection measure in a sample of participants surveyed three times between February 2020 and May 2020, by correlating it with commonly used connection measures (e.g., positivity resonance), as well as with measures of related constructs (e.g., personality, well-being).

## Study 2

Our second set of studies (involving three timepoints, labeled Time 1, 2, and 3) aimed to test the psychometric properties of the Connection During Conversations Scale. We also correlated this new scale with other similar measures of social connection-relevant constructs—namely, loneliness, relatedness, partner responsiveness, shared reality, and positivity resonance—to establish construct validity.

## Method

**Participants.**    At Time 1, a new set of participants (*N* = 399) were recruited from Prolific in January/February 2020, with the same eligibility criteria and sample size reasoning as Study 1. We removed 2 participants because they reported being younger than 18, yielding a final sample of *N* = 397. Participants at Time 1 ranged in age from 18 to 76 (*M* = 31.59, *SD* = 11.87), with 55% male, 80% Caucasian, and 47% single. Most were from the U.S. (32%) and the U.K. (27%), with the remainder (41%) from 26 other countries (e.g., Ireland, Portugal, Canada). Participants who returned at Time 2 (*N* = 336; April 2020) and Time 3 (*N* = 299; May 2020) were re-recruited from Time 1 and thus showed almost identical demographics. Those at Time 2 ranged in age from 18 to 72 (*M* = 32.03, *SD* = 11.94), with 55% male, 80% Caucasian, and 45% single. They resided around the world, with 31% from the U.S., 27% from the U.K., and the remaining 42% of participants from 26 international countries. Participants at Time 3 ranged in age from 18 to 69 (*M* = 32.13, *SD* = 11.92), with 53% male, 81% Caucasian, and 43% mostly single, 28% from the U.S., 27% from the U.K., and the remaining 45% of participants from 25 international countries.

**Procedure.**    The procedures and surveys completed at Time 1, 2, and 3 were highly similar and were designed to assess test-retest reliability (or correlations among the CDCS and its sub-scales) across the three time points. At all three timepoints, participants were reimbursed $3.75 on Prolific for a study titled "A Social Interaction Psychological Research Survey," with their participation lasting 25, 19, and 20 mins, respectively. Following written consent online, participants first completed our prompt asking them to take a few moments to describe an interpersonal interaction that had taken place within the last 2 days, to ensure the interaction was fresh and cognitively accessible in their minds. Then participants completed our 16-item connection measure, followed by questions about their specific social interaction, their interaction partner, and demographic items about themselves. Participants at Time 1 completed our full set of measures (e.g., positivity resonance, loneliness, personality), while at Time 2 and 3, participants responded to a subset of these measures (outlined below). Although we expected that test-retest stability may be relatively low (due to the uniqueness of each social interaction and partner), this repeated assessment allowed us to examine the stability and consistency of the CDCS over time.

## Materials

In addition to various demographic and interaction specific variables, seven measures were used in Study 2. The sample means, standard deviations, and reliability coefficients (Cronbach's alphas and Mcdonald's omegas) for each measure are reported in Table 3.

**Interaction-specific measures.**    *Connection During Conversations Scale*. Participants were asked to respond to our 16-item measure of interaction-specific social connection developed in Study 1 on a 1 (*strongly disagree*) to 7 (*strongly agree*) Likert scale. These items, including those that were reverse coded in analyses, are shown in Table 2.

*Interaction partner demographics and interaction details*. Participants again reported the interaction partner demographics and interaction details from Study 1 (see Table 1).

*Partner Responsiveness*. The 12-item Partner Responsiveness Scale [39, 40], again completed about their interaction partner, contains items like ". . .understands me" and ". . .sees the 'real' me" (1 = *strongly disagree*, 7 = *strongly agree*).

*Positivity Resonance*. Participants completed the 7-item Positivity Resonance Scale about their specific interaction [16] (e.g., "Did you feel a sense of mutual trust with (your

interaction partner)?" and "Did thoughts and feelings flow with ease between you and your interaction partner?"). Responses were made as percentages of time spent on the social interaction, on a sliding 0 to 100 percent scale, where higher numbers indicated greater positivity resonance.

*Shared Reality*. Participants also responded to the 8-item Shared Reality Scale about the social interaction [47] (e.g., ". . .the way we thought became more similar" and ". . .we saw the world in the same way"), using a 1 (*strongly disagree*) to 7 (*strongly agree*) Likert scale.

**General measures.** *Relatedness*. Participants responded to the 6-item relatedness subscale of the BMPN [34], which has items such as "I felt a sense of contact with people who care for me, and whom I care for" and "I felt close and connected with other people who are important to me," rated on 1 (*strongly disagree*) to 7 (*strongly agree*) Likert scales.

*Loneliness*. Participants completed the 20-item UCLA Loneliness Scale [38]. Sample items include "No one really knows me well" and "My social relationships are superficial," rated on 1 (*never*) to 4 (*often*) Likert scales, with higher scores indicating greater loneliness.

*Personality*. Participants responded to the extraversion facet only (Time 1: $M = 2.90$, $SD = 0.78$, $\alpha = .87$; Time 2: $M = 3.86$, $SD = 1.09$, $\alpha = .89$; Time 3: $M = 3.91$, $SD = 1.10$, $\alpha = .89$) of the 60-item Big Five Inventory-2 [49] on 1 (*strongly disagree*) to 5 (*strongly agree*) scales.

## Results

**Confirmatory factor analysis.** We conducted CFAs at each of our three timepoints on our 16-item measure of connection to assess whether our four-factor solution was a good fit. All CFAs were conducted in R using the *cfa* function in the lavaan package, with maximum likelihood estimation applied. At time 1, the four-factor CFA fit our connection items well, $\chi^2(98) = 378.80$, CFI = .932, TLI = .916, RMSEA = .085, 90% CI [.076, .094], SRMR = .054. At Time 2, again, the four-factor solution was a good fit, $\chi^2(98) = 378.84$, CFI = .925, TLI = .908, RMSEA = .092, 90% CI [.083, .102], SRMR = .059. At Time 3, a four-factor CFA also fit our connection items well, $\chi^2(98) = 367.39$, CFI = .930, TLI = .915, RMSEA = .096, 90% CI [.086, .106], SRMR = .050.

We also conducted correlations among each of the latent variables for each of our three timepoints. The Shared Reality latent variable was strongly correlated with the Partner Responsiveness latent variable ($r = .86$), the Participant Interest latent variable ($r = .76$), and the Affective Experience latent variance ($r = -.78$). The Partner Responsiveness was also strongly correlated with the Participant Interest latent variable ($r = .77$) and the Affective Experience latent variable ($r = -.72$). Finally, the Participant Interest latent variable and the Affective Experience latent variable were also negatively correlated ($r = -.80$).

**Correlations among the Connection During Conversations Scale and other measures.** Table 3 displays representative correlations for participants at Time 1 between our Connection During Conversations Scale, its four subscales, and similar scales that measure social connection in the literature. First, as expected, our overall scale was highly correlated (*r*s ranging from .68 to .84) with the Positivity Resonance Scale, Partner Responsiveness Scale, and Shared Reality Scale (the latter two being reflected in two of the subscales in our measure) and moderately correlated ([*r*]s ranging from .25 to .34) with the relatedness subscale of the BMPN, loneliness, and extraversion. Again, as expected, the four subscales were highly correlated with one another, with *r*s ranging from .54 (between the Shared Reality subscale and Participant Interest subscale) to .84 (between the Shared Reality subscale and Partner Responsiveness subscale).

**Table 3. Correlations among the Connection During Conversations Scale (CDCS), its four subscales, and other relevant connection scales (Study 2).**

| | CDCS (1) | SR (2) | PR (3) | PI (4) | AE (5) | Extraversion (6) | Loneliness (7) | Relatedness (8) | Partner Responsive (9) | Shared Reality (10) | Positivity Resonance (11) |
|---|---|---|---|---|---|---|---|---|---|---|---|
| | | | | | | Study 2 Time 1 | | | | | |
| *Mean (SD)* | 5.40 (1.11) | 5.08 (1.40) | 5.41 (1.30) | 5.70 (1.03) | 5.39 (1.44) | 2.90 (.78) | 2.17 (.65) | 4.88 (1.10) | 5.37 (1.27) | 4.82 (1.21) | 70.73 (24.14) |
| *Alpha* | .93 | .90 | .91 | .73 | .80 | .87 | .94 | .76 | .97 | .94 | .96 |
| *Omega* | .95 | .91 | .92 | .80 | .81 | .90 | .95 | .88 | .97 | .95 | .97 |
| 1 | - | | | | | | | | | | |
| 2 | .89** | - | | | | | | | | | |
| 3 | .90** | .84** | - | | | | | | | | |
| 4 | .77** | .54** | .61** | - | | | | | | | |
| 5 | .85** | .64** | .63** | .58** | - | | | | | | |
| 6 | .18** | .11* | .16** | .18** | .17** | - | | | | | |
| 7 | -.25** | -.15** | -.25** | -.22** | -.23** | -.57** | - | | | | |
| 8 | .34** | .26** | .35** | .26** | .29** | .35** | -.70** | - | | | |
| 9 | .79** | .76** | .83** | .51** | .58** | .21** | -.29** | .39** | - | | |
| 10 | .68** | .76** | .68** | .40** | .47** | .15** | -.18** | .26** | .73** | - | |
| 11 | .84** | .79** | .79** | .57** | .70** | .17** | -.26** | .36** | .80** | .70** | - |
| | CDCS (1) | SR (2) | PR (3) | PI (4) | AE (5) | Extraversion (6) | Loneliness (7) | Relatedness (8) | | | |
| | | | | | | Study 2 Time 2 | | | | | |
| *Mean (SD)* | 5.51 (1.08) | 5.28 (1.41) | 5.54 (1.29) | 5.79 (.98) | 5.44 (1.38) | 3.86 (1.09) | 2.16 (.49) | 4.91 (1.14) | | | |
| *Alpha* | .93 | .92 | .91 | .75 | .79 | .89 | .94 | .77 | | | |
| *Omega* | .95 | .95 | .91 | .82 | .80 | .92 | .95 | .89 | | | |
| 1 | - | | | | | | | | | | |
| 2 | .89** | - | | | | | | | | | |
| 3 | .91** | .83** | - | | | | | | | | |
| 4 | .74** | .51** | .59** | - | | | | | | | |
| 5 | .84** | .64** | .83** | .54** | - | | | | | | |
| 6 | .12+ | .13+ | .07 | .14** | .06 | - | | | | | |
| 7 | -.33** | -.27** | .26** | -.32** | -.29** | -.51** | - | | | | |
| 8 | .36** | .27** | .31** | .33** | .34** | .29** | -.64** | - | | | |
| | CDCS (1) | SR (2) | PR (3) | PI (4) | AE (5) | Extraversion (6) | Loneliness (7) | Relatedness (8) | | | |
| | | | | | | Study 2 Time 3 | | | | | |
| *Mean (SD)* | 5.48 (1.17) | 5.19 (1.47) | 5.43 (1.34) | 5.80 (1.02) | 5.51 (1.44) | 3.91 (1.10) | 2.27 (.63) | 4.91 (1.16) | | | |
| *Alpha* | .95 | .93 | .91 | .81 | .83 | .89 | .93 | .80 | | | |
| *Omega* | .96 | .93 | .94 | .84 | .86 | .92 | .96 | .88 | | | |
| 1 | - | | | | | | | | | | |
| 2 | .92** | - | | | | | | | | | |
| 3 | .92** | .87** | - | | | | | | | | |
| 4 | .82** | .66** | .67** | - | | | | | | | |
| 5 | .86** | .69** | .68** | .64** | - | | | | | | |
| 6 | .26** | .20** | .20** | .22** | .28** | - | | | | | |
| 7 | -.33** | -.23** | -.29** | -.34** | -.32** | -.55** | - | | | | |
| 8 | .36** | .27** | .30** | .38** | .34** | .34** | -.71** | - | | | |

*Note*. SR = Shared Reality subscale. PR = Partner Responsiveness subscale. PI = Participant Interest subscale. AE = Affective Experience subscale. Study 3 used a 14-item version of the CDCS.

+$p < .05$.

*$p < .01$.

**$p < .001$.

When examining the correlation between the four subscales of our Connection During Conversations Scale and previous social connection measures, the correlations followed similar patterns to the overall scale. For example, our Shared Reality subscale was highly correlated with the Positivity Resonance Scale, Partner Responsiveness Scale, and Shared Reality Scale (*r*s ranging from .76 to .79) and relatively more weakly correlated with relatedness, loneliness, and extraversion (*r*s between .26 and -.15). The other three subscales followed a similar trend, revealing strong correlations with the Positivity Resonance Scale, Partner Responsiveness Scale, and Shared Reality Scale. See Table 3 for the full correlation matrix.

**Correlations among Study 2 timepoints 1, 2, and 3.** Table 4 displays correlations among each of the timepoints in Study 2 to examine correlates on the CDCS, its subscales, and related scales. Correlations of the CDCS from Time 1, 2, and 3 were all significant and moderate (*r*s ranging from .27–32). For item-level correlations see Supplemental Materials S2 Table.

**Table 4. Correlations among the Connection During Conversations Scale (CDCS), its four subscales, and other relevant connection scales, across three occasions (Times 1, 2, and 3) in Study 2.**

| | CDCS | SR | PR | PI | AE | Extraversion | Loneliness | Relatedness |
|---|---|---|---|---|---|---|---|---|
| **Correlations between Time 1 and Time 2** | | | | | | | | |
| CDCS | .31** | | | | | | | |
| SR | .24** | .20** | | | | | | |
| PR | .29** | .21** | .30** | | | | | |
| PI | .24** | .14+ | .22** | .28** | | | | |
| AE | .30** | .19** | .24** | .30** | .31** | | | |
| Extraversion | .14+ | .10 | .10 | .13+ | .15** | .89** | | |
| Loneliness | -.21** | -.13+ | -.17* | -.24** | -.20** | -.47** | .80** | |
| Relatedness | .18** | .10 | .13+ | .25** | .16** | .24** | -.47** | .50** |
| **Correlations between Time 1 and Time 3** | | | | | | | | |
| CDCS | .27** | | | | | | | |
| SR | .19** | .15** | | | | | | |
| PR | .19** | .14+ | .21** | | | | | |
| PI | .19** | .19** | .22** | .32** | | | | |
| AE | .28** | .21** | .26** | .27** | .32** | | | |
| Extraversion | .22** | .16* | .17** | .21** | .24** | .89** | | |
| Loneliness | -.29** | -.21** | -.25** | -.28** | -.29** | -.56** | .81** | |
| Relatedness | .20** | .12** | .17** | .23** | .19** | .33** | -.59** | **.55**** |
| **Correlations between Time 2 and Time 3** | | | | | | | | |
| CDCS | .32** | | | | | | | |
| SR | .27** | .28** | | | | | | |
| PR | .28** | .26** | .30** | | | | | |
| PI | .32** | .20** | .30** | .37** | | | | |
| AE | .26** | .19** | .21** | .19** | .30** | | | |
| Extraversion | .11 | .12+ | .07 | .10 | .07 | .92** | | |
| Loneliness | -.34** | -.25** | -.27** | -.30** | -.32** | -.49** | .87** | |
| Relatedness | .32** | .24** | .25** | .29** | .31** | .28** | -.59** | .61** |

*Note.* SR = Shared Reality subscale. PR = Partner Responsiveness subscale. PI = Participant Interest subscale. AE = Affective Experience subscale.

+*p* < .05.

* *p* < .01.

** *p* < .001.

## Study 3

Because two items in all three Study 2 timepoints (items 12 and 13 in Table 2) had factor loadings below .50, the generally accepted cutoff for newly developed items [50], we recruited a new sample to validate the CDCS without these two items.

### Method

**Participants.** In Study 3, a new set of participants (*N* = 235) were recruited from a medium-sized public university in the U.S. and were granted research credit for their participation. The study was approved by the University of California, Riverside Institutional Review Board, and participants provided written consent to the study online. Participants ranged in age from 18 to 40 (*M* = 19.82, *SD* = 2.02) and were slightly more female (58%), plurality Asian (42%), and majority never married (64%). Their parents' highest level of education was some college (25%) or a 4-year college (20%).

**Procedure.** Participants completed a 30-min survey online, which comprised the Connection During Conversations Scale, as well as some measures used in Study 2, as well as new measures (e.g., Satisfaction With Life Scale, BMPN), to further assess construct and discriminant validity. In this study, the participants were asked to recall and write about their social interaction, but they were not asked to rate the interaction or their partner. Participants also responded to items about the COVID-19 pandemic, but analysis of these items is beyond the scope of the present study.

### Materials

**Interaction-specific measures.** *Connection During Conversations Scale.* Participants were asked to respond to our reduced 14-item measure of interaction-specific social connection developed in Study 1. These items, including those that were reverse coded in all analyses, are shown in Table 5.

**General measures.** *Affect.* Participants responded to a modified 15-item version of the Affect Adjective Scale [51], which includes both high and low arousal positive affect (PA; e.g., joyful, peaceful/serene) and negative affect (NA; e.g., angry/hostile, dull/bored, embarrassed) that participants used to assess their affect in the past 7 days (PA: *M* = 4.14, *SD* = 1.17, α = .91; NA: *M* = 3.58, *SD* = 1.18, α = .85).

*Autonomy, competence, and relatedness.* Participants in Study 3 completed the full 18-item BMPN, using 1 (*strongly disagree*) to 7 (*strongly agree*) Likert scales, which included the autonomy (*M* = 4.19, *SD* = 0.62, α = .51), competence (*M* = 3.91, *SD* = 0.74, α = .71), and relatedness subscales (*M* = 4.37, *SD* = 0.75, α = .69).

*Loneliness.* Participants again responded to the UCLA Loneliness Scale (*M* = 2.08, *SD* = 0.56, α = .93).

*Life satisfaction.* The 5-item Satisfaction With Life Scale [52] includes items such as "I am satisfied with my life" (1 = *strongly disagree*, 7 = *strongly agree*; *M* = 4.06, *SD* = 1.29, α = .86).

*Personality.* Participants responded to the 60-item Big Five Inventory-2 for all five facets (Extraversion *M* = 3.09, *SD* = 0.69, α = .86; Conscientiousness *M* = 3.34, *SD* = .62, α = .84; Neuroticism *M* = 3.07, *SD* = 0.72, α = .86; Openness *M* = 3.58, *SD* = .62, α = .82; and Agreeableness *M* = 3.67, *SD* = .53, α = .77).

### Results

**Confirmatory factor analysis.** We conducted a CFA using the *cfa* function in the lavaan package in R using maximum likelihood estimation on our reduced 14-item measure of

**Table 5. Connection During Conversations Scale.**

| 1 | 2 | 3 | 4 | 5 | 6 | 7 |
|---|---|---|---|---|---|---|
| Strongly disagree | Disagree | Somewhat disagree | Neither agree nor disagree | Somewhat agree | Agree | Strongly agree |

Please answer the following questions about your recent interaction and interaction partner.

Shared Reality Subscale

 1. I felt "in sync" with them

 2. I felt like we shared a lot in common

 3. I felt that we saw the world in the same way

 4. They were able to relate to my experiences

Partner Responsiveness Subscale

 5. They were interested in my thoughts and feelings

 6. They respected my beliefs and opinions

 7. I felt that they cared about me

 8. They really understood who I am

Participant Interest Subscale

 9. I was truly attentive during the interaction

 10. I was interested in their thoughts and feelings

 11. I thought that they were boring (R)

Affective Experience Subscale

 12. I felt that my energy was drained by the interaction (R)

 13. I couldn't wait for the interaction to end (R)

 14. I felt that it was hard to communicate with them (R)

connection to assess whether our four-factor solution was a good fit. A four-factor CFA fit our connection items well, $\chi^2(71) = 149.360$, CFI = .949, TLI = .935, RMSEA = .069; 90%CI [.053, .084], SRMR = .045. Correlations among latent variables were strong. The Shared Reality latent variable was strongly correlated with the Partner Responsiveness latent variable ($r = .86$), the Participant Interest latent variable ($r = .76$), and the Affective Experience latent variance ($r = -.78$). The Partner Responsiveness was also strongly correlated with the Participant Interest latent variable ($r = .77$) and the Affective Experience latent variable ($r = -.72$). Finally, the Participant Interest latent variable and the Affective Experience latent variable were also negatively correlated ($r = -.80$).

**Correlations among the Connection During Conversations Scale and other measures.** Table 6 displays correlations between the CDCS, its four subscales, and the other social connection scales included in this study. These correlations slightly diverge from Study 2 because we removed two items—one item from the Participant Interest subscale and one item from the Affective Experience subscale. First, as expected, our scale overall was moderately correlated with the relatedness subscale of the BMPN ($r = .58$) and loneliness ($r = -.61$) but relatively more weakly correlated with extraversion ($r = .36$). The four subscales were also highly correlated with one another, with $r$s ranging from .40 (between Shared Reality and Affective Experience) to .80 (between Shared Reality and Partner Responsiveness).

When examining the associations between the four subscales of our Connection During Conversations Scale and similar scales that assess social connection in the literature, again the correlations replicated the patterns obtained with the full (now) 14-item measure. For example, the Partner Responsiveness subscale was moderately correlated with relatedness ($r = .51$) and loneliness ($r = -.58$) but more relatively weakly correlated with extraversion ($r = .32$). All other subscales followed a similar trend. See Table 6 for the full correlation matrix.

**Table 6. Correlations among the Connection During Conversations Scale (CDCS), its four subscales, and other relevant scales (Study 3).**

Study 3

| | CDCS (1) | SR (2) | PR (3) | PI (4) | AE (5) | Extraversion (6) | Loneliness (7) | Relatedness (8) | Autonomy (9) | Competence (10) | Life Satisfaction (11) | Positive Affect (12) | Negative Affect (13) | Neuroticism (14) | Agreeable (15) | Conscientious (16) | Open (17) |
|---|---|---|---|---|---|---|---|---|---|---|---|---|---|---|---|---|---|
| Mean (SD) | 5.13 (.95) | 5.11 (1.13) | 5.45 (1.02) | 5.27 (1.10) | 4.68 (1.35) | 3.09 (.69) | 2.08 (.56) | 3.63 (.53) | 4.19 (.62) | 3.91 (.74) | 4.06 (1.29) | 4.14 (1.17) | 3.58 (1.18) | 3.07 (.72) | 3.67 (.53) | 3.34 (.62) | 3.58 (.62) |
| Alpha | .92 | .88 | .89 | .75 | .75 | .86 | .93 | .69 | .51 | .71 | .86 | .91 | .85 | .86 | .77 | .84 | .82 |
| Omega | .94 | .91 | .92 | .80 | .82 | .90 | .94 | .82 | .59 | .87 | .87 | .93 | .91 | .89 | .86 | .90 | .84 |
| 1 | - | | | | | | | | | | | | | | | | |
| 2 | .84** | - | | | | | | | | | | | | | | | |
| 3 | .87** | .80** | - | | | | | | | | | | | | | | |
| 4 | .88** | .68** | .72** | - | | | | | | | | | | | | | |
| 5 | .77** | .40** | .47** | .58** | - | | | | | | | | | | | | |
| 6 | .36** | .32** | .32** | .23* | .32** | - | | | | | | | | | | | |
| 7 | -.61** | -.53** | -.58** | -.51** | -.44** | -.47** | - | | | | | | | | | | |
| 8 | .58** | .43** | .51** | .53*** | .48** | .21* | -.62* | - | | | | | | | | | |
| 9 | .31** | .18* | .32** | .29** | .26** | .26** | -.46** | .40** | - | | | | | | | | |
| 10 | .27** | .19** | .23** | .29** | .21** | .34** | -.51** | .37** | .44** | - | | | | | | | |
| 11 | .31** | .33** | .36** | .24* | .13+ | .17+ | -.46** | .38** | .15** | .36** | - | | | | | | |
| 12 | .37** | .37** | .41** | .34** | .15** | .26* | -.49** | .45** | .32** | .49** | .56** | - | | | | | |
| 13 | -.23** | -.08 | -.08 | -.24** | -.33** | -0.10 | .39** | -.43** | -.35** | -.46** | -.16** | -.23** | - | | | | |
| 14 | -.27** | -.14 | -.19+ | -.31** | -.25* | -0.16+ | .50** | -.42** | -.31** | -.54** | -.33** | -.38** | .56** | - | | | |
| 15 | .40** | .27** | .32** | .37** | .35** | .23** | -.37** | .31** | .31** | .25* | .16+ | .18+ | -.23** | -.20* | - | | |
| 16 | .21** | .06 | .16+ | .23** | .24* | .22** | -.35** | .23** | .32** | .49** | .26** | .21** | -.23** | -.36** | .39** | - | |
| 17 | .24* | .24* | .23* | .19+ | .14 | .28** | -.17+ | .15 | .17+ | .15 | .17+ | .16+ | -.03 | .06** | .20+ | .16** | - |

*Note.* SR = Shared Reality subscale. PR = Partner Responsiveness subscale. PI = Participant Interest subscale. AE = Affective Experience subscale. Study 3 used a 14-item version of the CDCS.

+*p* < .05.

\**p* < .01.

\*\**p* < .001.

**Regression analyses.** Because our subscales were highly inter-correlated ($r$s ranging from .40 to .80), we conducted a series of regression analyses where each of the various outcome variables was regressed on the four subscales of the CDCS. Indeed, we found that our four subscales uniquely predicted various outcomes. For example, only Partner Responsiveness significantly predicted life satisfaction ($b = .30$, $SE = .13$, $p = .014$) and general PA in the past 7 days ($b = .32$, $SE = .12$, $p = .011$), only Affective Experience significantly predicted general NA ($b = -.30$, $SE = .07$, $p < .001$), and only Shared Reality significantly predicted conscientiousness ($b = -.14$, $SE = .07$, $p = .034$). Additionally, both Partner Responsiveness and Affective Experience significantly predicted relatedness (Partner Responsiveness: $b = .17$, $SE = .07$, $p = .014$; Affective Experience: $b = .16$, $SE = .04$, $p < .001$) and loneliness (Partner Responsiveness: $b = -.16$, $SE = .05$, $p < .001$; Affective Experience: $b = -.09$, $SE = .03$, $p = .001$). Given the high correlations among the CDCS subscales, we also calculated the variance inflation factors (VIFs) for each subscale in the regression models. Since all VIFs fell below the commonly used threshold of 10 (VIFs ranged from 1.62 to 2.85), this suggests multicollinearity was not a major concern in our analyses [53]. Table 7 displays the full set of regression analyses. In sum, each of our four subscales, despite being highly correlated, uniquely predicted several positive and negative psychological outcomes.

## Discussion

By compiling and updating items from existing measures in the literature that assess different aspects of social connection and interpersonal relationships, we created a new 14-item measure of social connection felt in a specific social interaction. Across three studies, we documented the reliability and validity of the Connection During Conversations Scale in measuring social connection in different social interactions. Furthermore, in Study 3, we demonstrated the uniqueness of each of our four subscales in predicting different outcomes. For example, the Shared Reality subscale was uniquely associated with conscientiousness; the Affective Experience subscale was uniquely associated with autonomy and loneliness; and the Partner Responsiveness subscale was uniquely associated with life satisfaction and positive affect in the last 7 days. As such, should researchers wish to look at connection as a whole (all 14 items) or a specific facet of connection, our findings provide preliminary evidence that each piece of the CDCS may offer unique information about the conversation and about the respondent.

Our measure fills a gap in the literature, as few existing scales specifically target aspects of social connection experienced during a specific interaction. Both researchers and laypeople have long known that fulfilling relationships are vital for social connection and well-being. However, what are interpersonal relationships but arguably simply a series of joint experiences, interactions, and conversations? Thus, not surprisingly, emerging research demonstrates that happy and socially connected people report having relatively frequent interactions [32]. Accordingly, we hope the CDCS will allow researchers to advance understanding of the psychological causes, mechanisms, and consequences of the connection felt during specific interactions. Future work as such may be able to identify what makes a conversation feel connecting. As just one example, researchers could test whether the common social etiquette of "not talking about religion or politics" really is an outdated sentiment and, if not, to identify potential boundary conditions (e.g., conversation length or type of interaction partner) that impact when hot-button topics are (or are not) connecting.

Furthermore, our measure contributes to the literature in that it captures four important facets or ingredients of social connection: shared reality, partner responsiveness, participant interest, and affective (or negative) experience. An extensive literature has already detailed the

**Table 7. Results of regression analyses of each of the four subscales of the Connection During Conversations Scale (CDCS) predicting our primary outcomes (Study 3).**

| | Adj R$^2$ | b(SE) | 95% CI | β | t | p |
|---|---|---|---|---|---|---|
| Life Satisfaction | .11 | | | | | |
| Shared Reality | | .16 (.12) | [-.07, .39] | .14 | 1.39 | .167 |
| **Partner Responsiveness** | | **.34 (.13)** | **[.07, .60]** | **.26** | **2.49** | **.014** |
| Participant Interest | | -.07 (.11) | [-.29, .15] | -.06 | -.64 | .525 |
| Affective Experience | | .01 (.07) | [-.14, .15] | .01 | .10 | .918 |
| Positive Affect (Last 7 days) | .19 | | | | | |
| Shared Reality | | .10 (.10) | [-.10, .29] | .09 | .98 | .330 |
| **Partner Responsiveness** | | **.30 (.12)** | **[.07, .53]** | **.26** | **2.58** | **.011** |
| Participant Interest | | .17 (.10) | [-.02, .36] | .16 | 1.75 | .081 |
| Affective Experience | | -.02 (.06) | [-.15, .10] | -.03 | -.35 | .724 |
| Negative Affect (Last 7 days) | .13 | | | | | |
| Shared Reality | | .01 (.10) | [-.20, .22] | .01 | .09 | .925 |
| Partner Responsiveness | | .15 (.12) | [-.09, .39] | .13 | 1.24 | .215 |
| Participant Interest | | -.15 (.10) | [-.35, .05] | -.14 | -1.46 | .146 |
| **Affective Experience** | | **-.30 (.07)** | **[-.43, -.17]** | **-.35** | **-4.59** | **< .001** |
| Relatedness (BMPN) | .35 | | | | | |
| Shared Reality | | .06 (.06) | [-.06, .17] | .08 | .97 | .332 |
| **Partner Responsiveness** | | **.17 (.07)** | **[.03, .30]** | **.22** | **2.49** | **.014** |
| Participant Interest | | .09 (.06) | [-.02, .20] | .13 | 1.55 | .212 |
| **Affective Experience** | | **.16 (.04)** | **[.09, .23]** | **.29** | **4.49** | **< .001** |
| Autonomy (BMPN) | .12 | | | | | |
| Shared Reality | | -.07 (.05) | [-.18, .03] | -.13 | -1.37 | .172 |
| **Partner Responsiveness** | | **.24 (.06)** | **[.12, .37]** | **.40** | **3.93** | **< .001** |
| Participant Interest | | .00 (.05) | [-.11, .10] | -.01 | -.08 | .935 |
| **Affective Experience** | | **.09 (.03)** | **[.02, .15]** | **.19** | **2.51** | **.012** |
| Competence (BMPN) | .07 | | | | | |
| Shared Reality | | .06 (.07) | [-.07, .20] | .10 | .97 | .336 |
| Partner Responsiveness | | .05 (.08) | [-.11, .20] | .06 | .60 | .550 |
| Participant Interest | | .03 (.07) | [-.10, .16] | .04 | .42 | .676 |
| Affective Experience | | .08 (.04) | [-.01, .16] | .14 | 1.79 | .074 |
| Loneliness | .38 | | | | | |
| Shared Reality | | -.08 (.04) | [-.16, .00] | -.16 | -1.86 | .064 |
| **Partner Responsiveness** | | **-.16 (.05)** | **[-.26, .04]** | **-.30** | **-3.40** | **< .001** |
| Participant Interest | | -.04 (.04) | [-.12, .04] | -.07 | -.94 | .346 |
| **Affective Experience** | | **-.09 (.03)** | **[-.14, -.03]** | **-.21** | **-3.27** | **.001** |
| Extraversion | .14 | | | | | |
| Shared Reality | | .12 (.07) | [-.02, .26] | .20 | 1.67 | .096 |
| Partner Responsiveness | | .10 (.08) | [-.06, .27] | .16 | 1.21 | .227 |
| Participant Interest | | -11 (.07) | [-.25, .04] | -.17 | -1.49 | .139 |
| **Affective Experience** | | **.13 (.04)** | **[.05, .21]** | **.27** | **3.04** | **.003** |
| Neuroticism | .09 | | | | | |
| Shared Reality | | .08 (.08) | [-.07, .24] | .13 | 1.06 | .290 |
| Partner Responsiveness | | -.01 (.09) | [-.19, .17] | -.02 | -.11 | .910 |
| **Participant Interest** | | **-.22 (.08)** | **[-.38, -.06]** | **-.33** | **2.75** | **.007** |
| Affective Experience | | -.05 (.05) | [-.14, .04] | .10 | -1.15 | .254 |
| Agreeable | .15 | | | | | |

*(Continued)*

**Table 7.** (Continued)

|  | Adj R² | b(SE) | 95% CI | β | t | p |
|---|---|---|---|---|---|---|
| Shared Reality |  | -.01 (.06) | [-.07, .24] | -.02 | -.21 | .837 |
| Partner Responsiveness |  | .06 (.06) | [.19, .17] | .12 | .95 | .343 |
| Participant Interest |  | .09 (.06) | [-.38, -.06] | .18 | 1.59 | .114 |
| **Affective Experience** |  | **.08 (.03)** | **[-.14, .04]** | **.20** | **2.33** | **.021** |
| Openness | .04 |  |  |  |  |  |
| Shared Reality |  | .08 (.07) | [-.06, .21] | .14 | 1.09 | .278 |
| Partner Responsiveness |  | .07 (.08) | [-.09, .23] | .12 | .90 | .369 |
| Participant Interest |  | .00 (.07) | [-.14, .13] | -.01 | -.06 | .949 |
| Affective Experience |  | .01 (.04) | [-.07, .10] | .03 | .36 | .717 |
| Conscientious | .07 |  |  |  |  |  |
| **Shared Reality** |  | **-.14 (.07)** | **[-.28, -.01]** | **-.27** | **-2.14** | **.034** |
| Partner Responsiveness |  | .10 (.08) | [-.06, .25] | .16 | 1.24 | .212 |
| Participant Interest |  | .12 (.07) | [-.02, .25] | .20 | 1.72 | .087 |
| Affective Experience |  | .07 (.04) | [-.01, .15] | .15 | 1.68 | .095 |

*Note.* One item in the Participant Interest subscale and all items in the Affective Experience subscale have been reverse coded. As such, positive values in Affective Experience indicate a positive experience. Variance inflation factors (VIFs) for each independent variable were as follows: Shared Reality VIF = 2.65; Partner Responsiveness VIF = 2.85, Participant Interest VIF = 2.13; Negative Experience VIF = 1.62.

critical role that the experience of shared reality and partner responsiveness play in a sense of overall social connection [13, 54]. That is, it is not surprising that two individuals who feel a commonality between one another (shared reality) or feel especially understood and valued by their partner (partner responsiveness) would report a strong sense of connection and a high-quality relationship.

Based on the regression analyses in Study 3, we have preliminary evidence demonstrating that participant interest and affective experience may also be important for various psychological outcomes, such as neuroticism and negative affect, respectively. That is, perhaps some of the items in the CDCS that specifically tap into a person's subjective experience during the interaction may be related to their personality and emotional state. Indeed, past research has shown that neurotic individuals often focus on the negatives and report relatively worse relationship satisfaction; our measure appears to pick up on this well-established phenomenon [55]. However, this study did not explicitly test this connection, but rather, the results presented provide preliminary evidence for such a phenomenon. Nonetheless, future research may benefit from aggregating multiple CDCS scores over time to see if this phenomenon holds true. Accordingly, our four subscales may provide meaningful insight into a variety of psychological outcomes.

In Study 2, we found that the 16-item version of the Connection During Conversations Scale was highly correlated with both existing conversation-specific measures of connection—namely, the Shared Reality ($r = .68$) and Positivity Resonance scales ($r = .84$). Although these correlations are high, our measure is different in a few key ways. First, the CDCS comprises three additional subscales beyond shared reality. Second, because our measure was not motivated by positivity resonance theory, it aims to assess social connection both as a broader and more comprehensive construct (i.e., the average of all items) and as tapping into four critical but separate ingredients of connection (i.e., the individual subscales of shared reality, partner responsiveness, participant interest, and affective experience). Furthermore, the CDCS can be

used to measure each of these features not only individually but in combination with one or two others (e.g., affective experience and participant interest but not partner responsiveness or shared reality). Such analyses may lead to unexpected insights—for example, what types of relationships, partners, or circumstances give rise to conversations that are interesting and engaging but do not lead one to feel in sync, valued, and understood? As such, our measure is not aligned with a specific theory of connection or limited to one feature of connection, but rather can tap into one to four critical ingredients of a connecting interaction depending on the research question.

## Limitations

A few limitations need to be addressed. First, the CDCS along with all other measures used across our three studies rely on self-reported data. This is a concern because we may see inflated relationships due to common method variance [56] or overly positive responses due to self-enhancement biases [57–59], the latter which is a problem for any socially desirable questionnaire such as those that measure happiness or life satisfaction [59, 60]. Although we did not assess common method variance in our statistical analyses, we employed various study design and data collection strategies to mitigate its potential impact, such as collecting multiple samples, providing clear instructions, and ensuring participant anonymity [61]. Second, the CFAs in Study 2 showed slightly elevated RMSEA values, which may raise concerns about model fit. It is important to note, however, that the other fit indices (CFI, TLI, and SRMR) demonstrated a good fit for the model, and the RMSEA is known to be a sensitive model that may overestimate lack of model fit [62]. Next, the sample sizes and composition of our samples, while relatively diverse in age (ranging from 18 to 70s), relationship status, and spanning countries around the world (e.g., the U.S., the U.K., Germany), were insufficient to make fine grained and complex comparisons. For example, our samples were too small to examine interactions between participant ethnicity and type of partner. Additionally, the samples recruited for Study 1 and 2 (predominantly White, male, internationally-based adults) differed substantially from the sample recruited for Study 3 (predominantly Asian, female, U.S. college students)— making specific comparisons more complex and difficult. Future investigators could oversample particular demographics or types of conversations and conversation partners in order to test comparisons and interaction effects. Another limitation is that our measure is designed to apply only to dyadic interactions—that is, to conversations between two individuals rather than groups of three or more. Of course, many conversations and social interactions —whether at a dinner party or Zoom brainstorming meeting—occur in a group or team context. Although not validated or intended to be used in this way, future studies could administer the CDCS multiple times (e.g., about Person A, B, and C) to assess felt social connection felt in a group conversation or adapt the instructions to refer to the group (e.g., whether one felt in sync with the group versus with a particular person).

## Future directions

Although we have outlined a few ideas for future directions above, there are further ways in which the CDCS can benefit future theory and research. Future investigators could bolster the generalizability of the CDCS by asking respondents to rate conversations with particular (and relatively infrequent) interaction partners, such as strangers, distant family members, and coworkers, or, alternatively, target long-term committed relationship partners. This approach may help to further establish the validity and reliability of our new scale within different types of relationships. However, as mentioned earlier, it is important to note that, when comparing scores on the CDCS for a single participant across several conversations (and conversation

partners), test-retest stability is not likely to be high, because each social interaction is expected to be unique. Regardless, we did find moderate correlations across three points in time on the connection measure in Study 2 (see Table 6).

Additionally, the CDCS can be used to assess whether certain types of interactions are more connecting than others. To address this question, researchers can focus on different aspects of conversations, such as interactions among specific types of interaction partners (e.g., family versus strangers, same-sex versus opposite sex, same versus different ethnicities, younger versus older dyads), the mode of communication (e.g., phone versus video), and the length of the conversation predicting feelings of connection. Relatedly, the CDCS can help identify which individual characteristics (e.g., personality, religious beliefs, political orientations) or conversation topics (e.g., personal stories, shared opinions, gossip) that make for more or less connecting moments. The results from such studies may help researchers identify both rifts and pinnacles of felt social connection and, thereby, to develop tools to repair or strengthen connecting moments in dyadic conversations.

We also recommend investigating alternative models, such as the bifactor model, to enhance understanding of the scale's underlying dimensions [63]. For example, do the four factors assess an essentially unidimensional construct of social connection? A bifactor approach could potentially reveal a general factor alongside specific factors, offering a more comprehensive perspective on the scale's internal structure and its relationship with other constructs.

Future investigators could also leverage a number of different methodologies in using the CDCS in studying human social interactions. For example, daily diary studies could examine how repeated interactions with the same person over time might predict feelings of connection. Furthermore, in experimental studies, participants could be instructed to have different types of conversations—for example, with a stranger who is matched versus mismatched on the Big Five; after a joy versus sadness mood induction, and face-to-face versus on video. Such studies would give researchers the opportunity to compare differences in the features or quality of connection experiences, as measured by the CDCS, after conversations with different types of partners, under different conditions, and using different modes of communication. For example, feelings of shared reality may be stronger for those conversing face-to-face than virtually because of the shared physical space, while negative affective experience may be higher for virtual conversations, due to awkwardness felt when someone is frozen or lagging. Notably, using the CDCS in face-to-face laboratory studies may also allow researchers to code nonverbal behaviors (e.g., leaning towards partner, arms crossed, fidgeting) during the conversations to add another dimension to help assess the quantity and quality of connection felt in conversation. Additionally, researchers could use the CDCS as part of ecologic momentary assessment to track, in real time, whether people are engaging in a conversation and, in that moment, how connected they are feeling. Such ratings could then be compared to the participants' retrospective self-reports (i.e., using the CDCS to rate the conversation at end of day or next day); differences between the "real-time" and retrospective reports could tap into social cognitive aspects of social connection.

Importantly, the CDCS may be valuable to investigate the antecedents, causes, mechanisms, and consequences of felt social connection. For example, by comparing different types of dyads (e.g., mother-daughter versus mother-son) that vary in closeness (e.g., interact daily versus monthly), mode of interaction (e.g., in person versus phone), conversation starting point (e.g., small talk vs. deep talk), and conversation topics (e.g., small talk versus problem solving versus reminiscing), future investigators may be able to disentangle which conversation features foster felt connection (e.g., begin with genuine interest), which maintain connection (e.g., shared memories), and which predict particular facets of connection, like partner responsiveness (e.g., in person conversations).

## Conclusion

An individual's overall sense of closeness, connection, and belonging is arguably derived from multiple conversations or social interactions—not only with partners, family members, and friends but with coworkers, acquaintances, and strangers. Because extensive research has shown that connection is vital for both mental and physical well-being [7], it is imperative for researchers to better understand how, when, where, and with whom people experience moments of connection in conversations. To this end, using a bottom-up approach, we developed our new Connection During Conversations Scale (CDCS), comprising four key facets of connection. The CDCS joins a very short list of measures that tap social connection felt during such specific conversations and interactions. We hope that this measure will allow researchers to identify what factors are associated with and promote the most connecting conversations in all kinds of dyads (including those diverging in closeness, personality, or political values) and in all kinds of circumstances (including conversations that are rushed, virtual, or glitchy). Ultimately, this work aims to inform future interventions that could both boost overall feelings of connection and help people connect across divides during specific social interactions.

## Supporting information

**S1 Table. List of connection-relevant scales used in scale creation and other recent scales.** (DOCX)

**S2 Table. Correlations among the Connection During Conversations Scale (CDCS) items across three occasions (Time 1, 2, and 3) in Study 2.** *Note.* SR = Shared Reality Subscale. PR = Partner Responsiveness Subscale. PI = Participant Interest Subscale. AE = Affective Experience subscale. $* p < .05.$ $** p < .01.$ $*** p < .001.$
(DOCX)

## Author Contributions

**Conceptualization:** Sonja Lyubomirsky.

**Formal analysis:** Karynna Okabe-Miyamoto, Lisa C. Walsh.

**Investigation:** Karynna Okabe-Miyamoto.

**Methodology:** Karynna Okabe-Miyamoto, Daniel J. Ozer, Sonja Lyubomirsky.

**Project administration:** Karynna Okabe-Miyamoto.

**Supervision:** Sonja Lyubomirsky.

**Writing – original draft:** Karynna Okabe-Miyamoto.

**Writing – review & editing:** Karynna Okabe-Miyamoto, Lisa C. Walsh, Daniel J. Ozer, Sonja Lyubomirsky.

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
