## [Decision Letter · Decision Letter 0]

13 Feb 2023

PONE-D-22-27914Measuring the Experience of Social Connection Within Specific Social Interactions: The Connection During Conversations Scale (CDCS)PLOS ONE

Dear Dr. Okabe-Miyamoto,

Thank you for submitting your manuscript to PLOS ONE. After careful consideration, we feel that it has merit but does not fully meet PLOS ONE’s publication criteria as it currently stands. Therefore, we invite you to submit a revised version of the manuscript that addresses the points raised during the review process.

We look forward to receiving your revised manuscript.

Kind regards,

Juan Antonio García, Ph.D.

Academic Editor

PLOS ONE

Journal Requirements:

2. Please change "caucasian” to "white” or "European" as appropriate (see for instance https://apastyle.apa.org/style-grammar-guidelines/bias-free-language/racial-ethnic-minorities)."

3. Please provide additional details regarding ethical approval in the body of your manuscript. In the Methods section, please ensure that you have specified the name of the IRB/ethics committee that approved your study.

4. Please provide additional details regarding participant consent. In the ethics statement in the Methods and online submission information, please ensure that you have specified what type you obtained (for instance, written or verbal, and if verbal, how it was documented and witnessed). If your study included minors, state whether you obtained consent from parents or guardians. If the need for consent was waived by the ethics committee, please include this information.

“NO”

7. We note that you have referenced (Karcher et al. ) which has currently not yet been accepted for publication. Please remove this from your References and amend this to state in the body of your manuscript: (Karcher et al. [Unpublished]”) as detailed online in our guide for authors http://journals.plos.org/plosone/s/submission-guidelines#loc-reference-style

Additional Editor Comments (if provided):

Dear authors,

I agreed to be reassigned as associate editor (AE) of this paper which was submitted to Plos One in October 2022. Given the difficulties encountered by the previous EA in finding two reviewers, I have opted to act as reviewer 1. I identify myself in this editorial letter to follow Plos One editorial policy. I hope that the two reviews below will be helpful to the authors.

Best regards,

Reviewers' comments:

Reviewer's Responses to Questions

**Comments to the Author**

1. Is the manuscript technically sound, and do the data support the conclusions?

Reviewer #1: Yes

Reviewer #2: Partly

2. Has the statistical analysis been performed appropriately and rigorously? 

Reviewer #1: No

Reviewer #2: Yes

3. Have the authors made all data underlying the findings in their manuscript fully available?

Reviewer #1: No

Reviewer #2: Yes

4. Is the manuscript presented in an intelligible fashion and written in standard English?

Reviewer #1: Yes

Reviewer #2: Yes

5. Review Comments to the Author

Reviewer #1: I enjoyed reading the paper and would like to congratulate the authors for the research presented. I think the paper is interesting and the contribution of the Connection During Conversations Scale (CDCS) is clearly justified. All three studies have an adequate sample size, but I would like to see the following changes in the statistical analyses presented:

- In study 1, an exploratory factor analysis is carried out with 53 items and then a confirmatory factor analysis with the 16 items but using the same sample. This is not a recommendable practice. Since this is a pilot study, I think it would be better to estimate two EFAs (with the 53 and 16 items, respectively), indicate the estimation method used, present the rotated factor loadings in both cases, and add the fit of both models (the psych package provides different indicators of model fit for the EFA).

- In study 2, I would like the authors to: (a) indicate the estimation method used for the CFA. Furthermore, it is observed that at all three time points (Time 1-3), the RMSEA is above 0.08. This indicates that the model fit is not adequate and may be because they have used an estimation method that is not the most appropriate; and (b) incorporate the assessment of stability at both item- and scale-level proposed by Dolnicar et al. (2022).

Dolnicar, S., Grün, B., & MacInnes, S. (2022). Assessing survey response stability: A complementary quality assurance protocol for survey studies in the social sciences. Social Sciences & Humanities Open,6(1), 100339. https://doi.org/10.1016/j.ssaho.2022.100339

- In study 3, I would like the authors to: (1) indicate the estimation method used for the CFA; and (2) present a comparison of the model fit of the 1-factor (unidimensional), 4-correlated factors (proposed model) and bifactor model (with one general factor and four specific factors). The authors should take advantage of this study 3 to further investigate the internal structure of the scale. Furthermore, in the regression analysis in Table 7, I miss some indicator of multicollinearity (e.g., tolerance or variance inflation factors). Considering the high correlations between the CDCS dimensions (see Table 5), I am afraid that the fact that only one or two parameters are significant in the regression analysis is due to the strong correlations between the independent variables (CDCS dimensions).

- In studies 1-3, I would like to see: (1) how the authors have assessed the possible existence of common method variance (they mention it as a limitation on page 26, but it is not considered in the statistical analyses); and (2) to add in addition to alpha another reliability indicator (at least omega and GLB, both for the four dimensions and for the CDCS).

Minor comments:

- Table 1 is very extensive, I think it would be better included as an appendix or supplementary material.

- On page 13, line 2, there is a “1” in superscript, but I don't see any note about it.

- On page 16, line 10, the authors mention “Time 2 and 4”, I think there is a mistake, and it is “Time 2 and 3”.

- Reorder Tables 5 and 6 (Table 5 refers to study 3).

Reviewer #2: Introduction

The introduction should be modified. Authors do not clearly specify what they understand by social connection and facets. Additionally, to prove construct validity, the study 2 and the Study 3 measure variables such as Personality, however authors in the introduction do not mention this relation.

Study 1: Materials – Connection Scale Item Pool: authors mention ‘most relevant items’. They should clarify what they understand by that expression.

Results – In this section authors start talking about the 16 items of the CDCS and the four latent variables and their correlations. However, the reader can’t understand where and how these variables group and where they come from. Authors provide the explanation later; this section should be rewritten.

When mentioning the CFA there are some negative correlations, authors should specify that.

Study 2: When introducing this study, authors state ‘social connection relevant constructs’, this should be justified (maybe in the general introduction section).

Procedure: It remains unclear the reason why participants complete three times the survey. The subsection should be rewritten to make it easier for the reader.

Study 3: Participants – More information should be provided, e.g.: country. Also, it seems some participants are underage, if so, authors should have asked for parental consent and this is not mentioned in the manuscript.

Procedure – Authors state that they use a subset of measures as in Study 2, but they are not. In the Study 3 they add life satisfaction, full BMPN and full scale of the Big Five, please rewrite this.

Results – On subsection Correlations among the connection during conversations scale and other measures, there is a typo regarding the number of the table. This is not Table 4 but Table 5.

Discussion

Limitation - Authors should make clear that, apparently, samples not similar between Study 1 and Study in comparison with Study 3.

Future directions – A semicolon should be removed, and a full stop should be added before the word ‘However’

Tables: Table 4, 5, and 6 should be adjusted to paper size.

6. PLOS authors have the option to publish the peer review history of their article (what does this mean?). If published, this will include your full peer review and any attached files.

Reviewer #1: No

Reviewer #2: No

---

## [Decision Letter · Decision Letter 1]

16 May 2023

Measuring the Experience of Social Connection Within Specific Social Interactions: The Connection During Conversations Scale (CDCS)

PONE-D-22-27914R1

Dear Dr. Okabe-Miyamoto,

We’re pleased to inform you that your manuscript has been judged scientifically suitable for publication and will be formally accepted for publication once it meets all outstanding technical requirements.

Kind regards,

Juan Antonio García, Ph.D.

Academic Editor

PLOS ONE

Additional Editor Comments (optional):

Reviewers' comments:

Reviewer's Responses to Questions

**Comments to the Author**

1. If the authors have adequately addressed your comments raised in a previous round of review and you feel that this manuscript is now acceptable for publication, you may indicate that here to bypass the “Comments to the Author” section, enter your conflict of interest statement in the “Confidential to Editor” section, and submit your "Accept" recommendation.

Reviewer #1: All comments have been addressed

Reviewer #2: All comments have been addressed

2. Is the manuscript technically sound, and do the data support the conclusions?

Reviewer #1: Yes

Reviewer #2: Yes

3. Has the statistical analysis been performed appropriately and rigorously? 

Reviewer #1: Yes

Reviewer #2: Yes

4. Have the authors made all data underlying the findings in their manuscript fully available?

Reviewer #1: Yes

Reviewer #2: Yes

5. Is the manuscript presented in an intelligible fashion and written in standard English?

Reviewer #1: Yes

Reviewer #2: Yes

6. Review Comments to the Author

Reviewer #1: The authors have paid attention to all my comments. The paper is ready for publication. Congratulations!

Reviewer #2: After reviewing the manuscript again, I consider this is an interesting paper that meets the journal's requirements to be published. The authors have addressed all the suggested changes, therefore the editor might consider its publication.

7. PLOS authors have the option to publish the peer review history of their article (what does this mean?). If published, this will include your full peer review and any attached files.

Reviewer #1: No

Reviewer #2: No

---

## [Editor Report · Acceptance letter]

22 Jun 2023

PONE-D-22-27914R1 

Measuring the experience of social connection within specific social interactions:
The Connection During Conversations Scale (CDCS) 

Dear Dr. Okabe-Miyamoto:

I'm pleased to inform you that your manuscript has been deemed suitable for publication in PLOS ONE. Congratulations! Your manuscript is now with our production department. 

Kind regards, 

on behalf of

Dr. Juan Antonio García 

Academic Editor

PLOS ONE